# Polyoxometalate-Modified Amphiphilic Polystyrene-*block*-poly(2-(dimethylamino)ethyl methacrylate) Membranes for Heterogeneous Glucose to Formic Acid Methyl Ester Oxidation

**DOI:** 10.3390/nano13182498

**Published:** 2023-09-05

**Authors:** Yurii Utievskyi, Christof Neumann, Julia Sindlinger, Konstantin Schutjajew, Martin Oschatz, Andrey Turchanin, Nico Ueberschaar, Felix H. Schacher

**Affiliations:** 1Institute of Organic Chemistry and Macromolecular Chemistry, Friedrich Schiller University Jena, Humboldtstraße 10, 07743 Jena, Germany; 2Institute of Physical Chemistry, Friedrich Schiller University Jena, Lessingstraße 10, 07743 Jena, Germany; 3Mass Spectrometry Platform, Faculty of Chemistry and Earth Sciences, Friedrich Schiller University Jena, Humboldtstraße 8, 07743 Jena, Germany; 4Institute for Technical Chemistry and Environmental Chemistry (ITUC), Friedrich Schiller University Jena, Philosophenweg 7a, 07743 Jena, Germany; 5Center for Energy and Environmental Chemistry (CEEC), Friedrich Schiller University Jena, 07743 Jena, Germany; 6Jena Center for Soft Matter (JCSM), Friedrich Schiller University Jena, 07743 Jena, Germany

**Keywords:** heterogeneous catalysis, glucose oxidation, polymer membranes, polyoxometalates

## Abstract

Herein, we present a new heterogeneous catalyst active toward glucose to formic acid methyl ester oxidation. The catalyst was fabricated via electrostatic immobilization of the inorganic polyoxometalate HPA-5 catalyst H_8_[PMo_7_V_5_O_40_] onto the pore surface of amphiphilic block copolymer membranes prepared via non-solvent-induced phase separation (NIPS). The catalyst immobilization was achieved via wet impregnation due to strong coulombic interactions between protonated tertiary amino groups of the polar poly(2-(dimethylamino)ethyl methacrylate) block and the anionic catalyst. Overall, three sets of five consecutive catalytic cycles were performed in an autoclave under 90 °С and 11.5 bar air pressure in methanol, and the corresponding yields of formic acid methyl ester were quantified via head-space gas chromatography. The obtained results demonstrate that the membrane maintains its catalytic activity over multiple cycles, resulting in high to moderate yields in comparison to a homogeneous catalytic system. Nevertheless, presumably due to leaching, the catalytic activity declines over five catalytic cycles. The morphological and chemical changes of the membrane during the prolonged catalysis under harsh conditions were examined in detail using different analytic tools, and it seems that the underlying block copolymer is not affected by the catalytic process.

## 1. Introduction

Amphiphilic block copolymers are frequently used as building blocks for the fabrication of asymmetric porous polymer membranes with functionalized pore surface via non-solvent-induced phase separation (NIPS) [1]. In this approach, the block copolymer solution is at first cast onto a polished glass substrate and then immersed into a coagulation bath filled with the non-solvent. The solvent for the amphiphilic block copolymer solution is often a THF/DMF mixture, while the non-solvent is water. Immersion into the coagulation bath leads to the solvent–non-solvent exchange, resulting in the formation of polymer-poor and polymer-rich phases. The polymer-poor phase eventually forms the pores, while the polymer-rich phase forms the membrane matrix [2,3]. Various block combinations have been used for membrane preparation via NIPS processes so far [1,2,3,4,5,6,7,8,9,10]. During the membrane formation process, the interaction of the hydrophilic block with water is more energetically favorable, which leads to the pore surface being covered with the hydrophilic segment after membrane formation [2,3]. This provides us with an opportunity to modify the membrane pore surface by synthesizing block copolymers with hydrophilic blocks bearing desired functional groups, which can then be used for further material modification for different applications. Recent examples include quaternization and cross-linking of the 4-vinylpyridine moieties of the PS-*b*-P4VP membrane, thereby providing permanent positive charge to the pore surface and at the same time enabling tuning of the effective pore size. The obtained membranes demonstrated the capability of separating small positively charged organic molecules in aqueous solution, which can then be applied in the areas of pollutant removal or chemical and pharmaceutical separations [11,12]. Another example includes the esterification of hydroxyl groups of polystyrene-block-poly(2-hydroxyethyl methacrylate) (PS-*b*-PHEMA) block copolymer with glycine, resulting in functionalization of the pore surface with primary amino groups. The latter enabled the successful covalent incorporation of fluorescein isocyanate and ninhydrin onto the membrane, thus providing it with sensing capabilities [13].

Over the past few years, our group developed heterogeneous catalytic systems utilizing different block copolymer membranes for the immobilization of inorganic catalysts: We reported on the immobilization of [PV_2_Mo_10_O_40_]^5−^ (HPA-2) to the surface of PS-*b*-PDMAEMA for oxidation catalysis [14] and [Mo_3_S_13_]^2−^ to the surface of (P(S-*co*-I)-*b*-PDMAEMA) membranes for hydrogen evolution reaction (HER) [15]. We could afterward successfully demonstrate sulfide to sulfone oxidation in the presence of hydrogen peroxide as an oxidant [14], as well as light-driven HER using cationic [Ru(bpy)_3_]^2+^ photosensitizer (PS) electrostatically attached to the [Mo_3_S_13_]^2−^ layer [15]. Further, we also demonstrated the successful covalent attachment of the Ru-based PS to the membrane made of poly(styrene-*co*-isoprene)-*block*-poly((4-vinylbenzyl chloride)-*co*-(tri(ethylene glycol) methyl ester acrylate)) (P(S-*co*-I)-*b*-P(VBCl-*co*-TEGA)) [16]. In our latest work, we successfully applied the PS-*b*-PDMAEMA membrane modified with electrostatically attached [Ru(bpy)_3_]^2+^ PS and polyoxometalate cluster [Co_4_(H_2_O)_2_-(PW_9_O_34_)_2_]^10−^ catalyst (POW-WOC) for water oxidation catalysis [17].

Currently, we aim to extend the scope of heterogeneous catalytic systems based on block copolymer membranes prepared via NIPS. We are particularly interested in recent work on the optimization of the homogeneous glucose to formic acid methyl ester oxidation under high molecular oxygen pressure—in this case, utilizing a H_8_[PV_5_Mo_7_O_40_] (HPA-5) phosphovanadomolybdate polyoxometalate (POM) catalyst [18].

This catalyst belongs to the class of phosphovanadomolybdates of the general formula H_3+x_[PV_x_Mo_12-x_O_40_], and such anions can be easily loaded onto a block copolymer membrane featuring positive surface charge [14]. Overall, the reactions in which phosphovanadomolybdates act as catalysts or co-catalysts can be divided into three main areas [19]. The first area includes oxidation reactions with dioxygen acting as a terminal oxidant; this type of reaction requires high oxygen partial pressures [19]. The second area is oxidation with the use of hydrogen peroxide [20,21], and the third one is water oxidation reaction [22,23,24]. 

By developing a new heterogeneous system in which H_8_[PV_5_Mo_7_O_40_], along with molecular oxygen, oxidizes glucose to formic acid methyl ester, we want to expand the scope of block copolymer membranes application in the area of phosphovanadomolybdate catalysis. In addition, the given heterogeneous system might be relevant for future production of a widely used industrial chemical, starting from biomass [25,26,27]. 

In this work, we present the first approach toward electrostatic immobilization of HPA-5 POM within the pores of amphiphilic PS-*b*-PDMAEMA block copolymer membranes. The resulting catalytically active membranes were characterized via various analytical techniques and tested for multiple consecutive cycles to catalyze glucose to formic acid methyl ester oxidation reactions at elevated temperature and high air pressure. The product yield was quantified using head-space gas chromatography, and we could demonstrate good reproducibility and comparable efficiency to the homogeneous system. In addition, the block copolymer membrane proved to be physically stable over prolonged reaction periods, and we did not detect significant oxidation of the underlying block copolymer.

## 2. Materials and Methods

### 2.1. Synthesis of the HPA-5 Heteropolyoxometalate

The heteropolyoxometalate catalyst H_8_[PV_5_Mo_7_O_40_] was prepared via a three-step synthesis [28]. In the first and second steps, 4.0 g (22 mmol) of vanadium oxide V_2_O_5_ was dissolved in water at 5 °C within 40 min, followed by dropwise addition of 32.5 mL (*ca.* 0.32 mol) of aqueous hydrogen peroxide solution (30% *w*/*w*) under vigorous stirring. Upon the complete addition of the hydrogen peroxide, the solution turned red, and the flask was removed from the ice bath and heated to 35 °C for 40 min. The reaction was accompanied by the formation of oxygen and darkening of the solution, identifying the formation of the H_6_[V_10_O_28_] intermediate. When the gas evolution was finished, 1.3 g (*ca*. 3.35 mmol) of phosphoric acid H_3_PO_4_ aqueous solution (25% *w*/*w*) was added slowly, and the mixture turned dark brown upon the formation of the H_9_PV_14_O_42_ precursor. The obtained solution was further used for the next step. In the third step, the molybdenum precursor was prepared by dispersing 8.85 g (61.5 mmol) of molybdenum oxide MoO_3_ in water under vigorous stirring, followed by addition of 2.1 g (*ca.* 5.4 mmol) of H_3_PO_4_ aqueous solution (25% *w*/*w*). The resulting mixture was heated up to 140 °C for 1 h, and the solution turned yellow (the color of the H_3_[PMo_12_O_40_] precursor). Afterward, the solution of the H_9_[PV_14_O_42_] precursor was gradually added to the boiling solution of the H_3_[PMo_12_O_40_] precursor, and the mixture was left to react for 3 h. The ratio of both precursors was stoichiometric in order to obtain the desired Mo/V ratio in the final product. The product solution was partially evaporated by rotary evaporation and then filtered to remove unreacted vanadium and molybdenum oxides. The filtrate was evaporated again, and the resulting brown crystalline compound was dried under vacuum overnight. The obtained yield was 13.5 g (96%). The obtained HPA-5 POM H_8_[PV_5_Mo_7_O_40_] catalyst was further analyzed via ^31^P, ^51^V NMR, TGA, XPS, as well as by ICP-OES. In the ^31^P spectrum, the signals of multiple positional isomers of the HPA-5 were observed in the interval from +1.0 to −4.0 ppm, as well as a free phosphate anion signal at +0.5 ppm. The ^51^V spectrum showed a group of broadened signals from −510 to −570 ppm, corresponding to multiple isomers of the HPA-5 catalyst. Both ^31^P and ^51^V spectra patterns were in good agreement with the previously published data [29]. The NMR data were supported by the results of the ICP-OES elemental analysis; the measured elemental ratio of P/Mo/V was P/Mo/V = 1/6,70/4,76, with the Mo/V ratio being 7/5, the latter being in exact agreement with HPA-5 H_8_[PV_5_Mo_7_O_40_] elemental composition. 

### 2.2. Synthesis of PS

The PS block was prepared via bulk polymerization of 16.5 mL (0.144 mol) of styrene at 110 °C for 24 h under argon atmosphere with the use of 55 mg (144 mmol) of the BlockBuilder MA nitroxide as an initiator. The prepared PS was precipitated in methanol and dried under high vacuum on a Schlenk line. The PS was characterized via size exclusion chromatography (SEC) using THF as an eluent and ^1^H NMR spectroscopy in CDCl_3_.

^1^H NMR (300 MHz, CDCl_3_, *δ* [ppm]): 7.24–6.26 (m, -C_6_H_5_), 2.30–1.10 (m, backbone -CH-CH_2_-).

SEC (THF, PS calibration): M_n_ = 76.0 kg/mol, M_w_ = 85.9 kg/mol, D = 1.13.

### 2.3. Synthesis of PS-b-PDMAEMA Block Copolymer

In the next step, 1.5 g (19.7 µmol) of the PS macroinitiator and 1.5 mL (8.8 mmol) of the DMAEMA monomer were dissolved in THF, and the reaction mixture was heated at 110 °C for 2 h under argon atmosphere. The resulting block-copolymer polymer was precipitated in hexane and dried afterward on a freeze dryer. The PS-*b*-PDMAEMA block copolymer was characterized via size exclusion chromatography (SEC) using THF as an eluent and ^1^H NMR spectroscopy in CDCl_3_.

^1^H NMR (300 MHz, CDCl_3_, *δ*): 7.24–6.26 (m, -C_6_H_5_), 4.20–3.95 (m, -N-CH_2_-), 2.69–2.53 (m, -O-CH_2_-), 2.40–2.20 (m, -N-CH_3_-), 2.15–0.75 (m, backbone -CH-CH_2_-, -C-CH_3_).

SEC (THF, PS calibration): M_n_ = 79.7 kg/mol, M_w_ = 101.9 kg/mol, D = 1.28.

### 2.4. Preparation of the PS-b-PDMAEMA Porous Membranes

The porous PS-*b*-PDMAEMA block copolymer membranes were fabricated by applying the NIPS process. For membrane fabrication, a 15 wt% block copolymer THF/DMF (70/30 wt%) solution was cast onto a glass plate via a 200 µm gate height doctor blade in a climate chamber from PlasLabs (Lansing, MI, USA) at controlled temperature (22 °C) and humidity (50%). The cast solution was exposed to the air for 30 s and then placed into a water bath for 10 s. Afterward, the polymer membrane formed was taken out from the water bath and stored in deionized water [1,9,14].

### 2.5. Preparation of the Catalytically Active Membranes

Catalytically active membranes were prepared via wet impregnation of pristine polymer membranes into an aqueous HPA-5 solution. The membranes were put into a 15 mL HPA-5 water solution with catalyst concentration of 1 mg/mL for 24 h. Afterward, the membrane was taken out, quickly washed two times with distilled water and then placed into distilled water for another 24 h. After 24 h, the water was replaced one more time. 

### 2.6. Catalytic Heterogeneous Glucose to Formic Acid Methyl Ester Oxidation

Heterogeneous glucose to formic acid methyl ester oxidation was carried out in a stainless-steel high-pressure OLT-HP-50 reactor (Xiamen Crtop Machine Co., Ltd., Xiamen, China) equipped with a stirring motor, heater and a removable 50 mL volume PTFE vessel. The reactor was placed inside a protective box and directly connected to the air supply line. For each reaction, the given masses of catalytic membrane and glucose were placed inside a PTFE vessel with a magnet bar; a polymer mesh was placed between the membrane and the magnet bar to prevent physical damage to the membrane during the reaction. Afterward, 20 mL of methanol and 100 µL of pentane (internal standard) were added to the vessel. The solution was placed into the high-pressure reactor and quickly sealed. The air pressure of 8.5 bar was applied, and the solution was stirred for 30 min at room temperature. After the saturation of methanol solution with air, the pressure in the system was again adjusted to 8.5 bar after a little pressure decrease. In the next step, the reactor temperature was set up to 90 °С for 24 h with a pressure of 11.5 bar inside upon heating. After cooling down the reactor to room temperature, the pressure was released. The reactor was closed again, and the solution was stirred for 30 min in order to remove excessive air dissolved in methanol. Afterward, the pressure was released again. The reactor was opened, and solution aliquots were taken for head-space GC–MS analysis. Prior to the next catalytic cycle, the remaining methanol solution was completely removed from the reaction vessel, the membrane was washed two times with fresh methanol and left to dry overnight.

### 2.7. Instrumentation

#### 2.7.1. NMR

^1^H NMR measurements were performed on a Bruker Avance I 300 MHz spectrometer using CDCl_3_ as a solvent.

^31^P and ^51^V NMR measurements were performed on a Bruker Avance III 500 MHz spectrometer using D_2_O as a solvent.

#### 2.7.2. SEC

An Agilent 1260 Infinity system (Agilent, Waldbronn, Germany) equipped with a 1260 IsoPump (G1310B), a 1260 ALS (G1310B) autosampler and three consecutive PSS SDV, 5 µm, 8 × 300 mm columns was used for SEC measurements. THF was used as an eluent at a flow rate of 1 mL·min^−1^. The column oven was set to 30 °C, and signals were detected using a 1260 DAD VL (G1329B) and a 1260 RID (G1315D) detector. The system was calibrated using PSS polystyrene (266 to 2,520,000 g·mol^−1^) standards.

#### 2.7.3. TGA

Thermogravimetric analysis (TGA) was performed on a Perkin-Elmer-TGA 8000 (PerkinElmer Inc., Waltham, MA, USA) instrument under airflow (20 mL min^−1^) at a heating rate of 10 °C min^−1^.

#### 2.7.4. ICP

ICP-OES was performed on a simultaneous radial ICP-OES spectrometer 725ES (Agilent Technologies GmbH, Waldbronn, Germany) with CCD detector equipped with autosampler ASX 520 for liquid sample injection. 

#### 2.7.5. SEM

Scanning electron microscopy (SEM) was performed on the Sigma VP field-emission scanning electron microscope (Carl-Zeiss AG, Oberkochen, Germany) operating at 6 kV using an InLens detector (Carl-Zeiss AG, Oberkochen, Germany). The samples were coated with Pt (5 nm) using CCU-010 HV high vacuum sputter coater (Safematic GmbH, Zizers, Switzerland). 

#### 2.7.6. XPS

XPS measurements were performed using a UHV Multiprobe system (Scienta Omicron, Taunusstein, Germany) with a monochromatic X-ray source (Al K_α_) and an electron analyzer (Argus CU) with 0.6 eV energy resolution. Charge compensation during data acquisition was realized by an electron flood gun (NEK 150-SC, Staib, Langenbach, Germany) at 6 eV and 50 µA. The background was subtracted, and spectra were calibrated using the C 1s peak (284.6 eV) before undergoing fitting using Voigt functions (30:70). For quantitative analysis, the relative sensitivity factors 1.00 (C 1s), 2.93 (O 1s), 1.80 (N 1s), 5.62 (Mo 3d_5/2_), 6.37 (V 2p_3/2_) and 0.79 (P 2p_3/2_) were used. In order to distinguish between the overlapping N 1s and Mo 3p_3/2_ the Mo 3p_3/2_ and the Mo 3p_1/2_ signals at ~416 eV were fitted using a fixed intensity ratio of 2:1 due to spin–orbit splitting. The remaining signal was assigned to the N 1s signal and fitted accordingly.

#### 2.7.7. Nitrogen Physisorption

Nitrogen physisorption isotherms at 77 K were recorded on a Quantachrome (3P Instruments, Odelzhausen, Germany) Quadrasorb SI gas sorption analyzer. The sample was degassed under vacuum at approx. 10^−3^ mbar at room temperature for 20 h. The specific surface area was calculated using the multi-point BET method in a *p*/*p*_0_ range of 0.075 < *p*/*p*_0_ < 0.25. The total pore volume was determined using the Gurvich rule at *p*/*p*_0_ = 0.98.

#### 2.7.8. Head-Space GC–MS

A volume of 2 mL of the analyzed solutions was transferred to 20 mL HS glass vials and crimp-capped with a PTFE/Silicone cap (MACHEREY-NAGEL GmbH & Co., Ltd, KG, Düren, Germany). Samples were measured on a GC–MS system from Thermo Fisher Scientific GmbH (Bremen, Germany) consisting of a TRACE™ 2000 GC with an S/SL injector installed and a Polaris Q ion trap mass spectrometer. The scan range was set between 30 and 400 *m*/*z*. The transfer line temperature was set to 300 °C, the ion source temperature to 200 °C, and damping gas flow was set to 0.3 mL min^−1^ helium. Three micro scans were recorded at a maximum ion time of 25 ms. After the initial 2 min at 40 °C, the GC oven temperature was raised to 100 °C with 10 °C min^−1^ and then to 200 °C with 50 °C min^−1^. The injector was operated in split mode at 270 °C with a flow rate of 10 mL min^−1^; the split ratio was 10 and the column flow 1 mL min^−1^. As a column, Zebron ZB-1MS 60 m from Phenomenex (Aschaffenburg, Germany) was used, with 0.25 mm inner diameter and 1 µm film thickness. Prior to injection, the sample vials were incubated at 40 °C for 30 s by shaking in a CombiPAL autosampler (CTC Analytics AG, Zwingen, Switzerland). The syringe temperature was set to 50 °C, and analysis was started by injecting 251 µL of the head-space content.

For quantification, a calibration curve using pentane as internal standard (0.5% final concentration in methanol) and formic acid methyl ester in methanol (in a concentration range from 0.05% to 0.55% in 0.10% steps as triplicates) was created. Data integration was carried out using Thermo Xcalibur Qual Browser version 1.4 SR1. The retention time of pentane was 7.2 min and 6.0 min for formic acid methyl ester, respectively. The quantification ions were 71 *m*/*z* for pentane and 61 *m*/*z* for formic acid methyl ester, respectively. The general settings for peak detection were as follows: mass deviation: ±0.5 u, ICIS peak detection, smoothing points: 1, baseline window: 40, area noise factor: 5, peak noise factor: 10. Linear regression was performed with OriginPro 9.8.0.200. The calculations were executed using Microsoft Excel Professional Plus 2019.

#### 2.7.9. Chemicals

Vanadium(V) oxide (99.6%), molybdenum(Vl) oxide (99.5%), aqueous phosphoric acid solution (85% *w*/*w*) and H_2_O_2_ aqueous solution (30% *w*/*w*) for the synthesis of the HPA-5 (H_8_[PV_5_Mo_7_O_40_]) were purchased from Alfa Aesar (ThermoFisher GmbH, Kendel, Germany). The styrene (≥99%) and 2-(Dimethylamino)ethyl methacrylate (98%) (DMAEMA) monomers were purchased from Sigma-Aldrich (Sigma-Aldrich CHEMIE GmbH, Steinheim, Germany). Formic acid methyl ester (97%) was purchased from Sigma-Aldrich (Merck KGaA, Darmstadt, Germany). D-(+)-Glucose (≥99.5%) and HPLC grade methanol for catalytic reaction were obtained from Sigma-Aldrich (Sigma-Aldrich CHEMIE GmbH, Steinheim, Germany) and VWR Chemicals (VWR International, Fontenay-sous-Bois, France), respectively. 

## 3. Results and Discussion

### 3.1. Preparation of the PS-b-PDMAEMA Block Copolymer 

The polystyrene-*block*-poly-2-(*N*,*N*-dimethylaminoethyl methacrylate) (PS-*b*-PDMAEMA) block copolymer was successfully synthesized in two steps via nitroxide-mediated polymerization (NMP) using BlockBuilder MA as an initiator (Figure 1). In the first step, the polystyrene (PS) block was prepared in bulk with a dispersity index of 1.13 and an average molecular weight of 76 kg/mol. The subsequent block extension with DMAEMA was carried out in THF using an optimized PS to DMAEMA monomer ratio of 1 to 450. Successful block extension was confirmed via size exclusion chromatography (SEC) (Appendix A), and both molar and weight shares of the PDMAEMA block were calculated via ^1^H NMR spectroscopy (Appendix A), leading to S_71_D_29_^107^, where the subscripts represent wt% of the PS (S) and PDMAEMA blocks (D), and the superscript represents the molecular weight in kg/mol.

The given amphiphilic block copolymer can be used for the formation of self-supporting nano-porous membranes via the NIPS process. Hereby, the hydrophilic block will cover the pore surface, while the hydrophobic segments will form the membrane matrix [30]. Based on this, we chose DMAEMA as a polar block due to its amino group, which can be easily protonated in acidic solutions, thereby enabling subsequent immobilization of the negatively charged polyoxometalate catalyst utilizing attractive electrostatic interactions. At the same time, the hydrophobic PS block prevents membrane swelling in methanol, which will ensure membrane stability over prolonged reaction times. The weight share of the hydrophilic PDMAEMA block of up to 30 wt% was previously found to be optimal for porous membrane fabrication via NIPS [1,9,14]. At the same time, lower molar shares might lead to smaller catalyst load onto a given membrane. 

### 3.2. Preparation of Catalytically Active Membranes

PS-*b*-PDMAEMA membranes were prepared using previously published conditions [14]; the detailed procedure can be found in Section 2. After preparation, the membranes could be stored for months in deionized water. The PDMAEMA units present on the pore surface enable the electrostatic attachment of the HPA-5 catalyst, while the PS matrix renders the membrane stable in polar media. 

The morphology of the polymer membranes prepared via the NIPS process was analyzed via SEM (Figure 2). Membrane casting under the chosen conditions resulted in a macro-porous volume structure and a more fine-porous surface layer. The total pore volume and specific surface area of the PS-*b*-PDMAEMA membrane—calculated from the nitrogen physisorption isotherm—are 0.101 cm^3^g^−1^ and 36 m^2^g^−1^, respectively. The isotherm and the corresponding multi-point BET fit are given in ESI (Appendix A).

The HPA-5 polyoxometalate H_8_[PV_5_Mo_7_O_40_] was prepared according to the previously published three-step synthetic methodology using MoO_3_, V_2_O_5_ and H_3_PO_4_ precursors in stoichiometric amounts [28]. The synthetic scheme with stoichiometric coefficients is given in the Appendix A. The catalyst was characterized via ^31^P, ^51^V NMR, TGA and ICP-OES elemental analysis. The ^31^P and ^51^V NMR spectra of HPA-5 measured in D_2_O (Appendix A) contain signals of multiple HPA-5 positional isomers, which increase in number upon the increase in the degree of Mo substitution with V. The signal patterns in ^31^P and ^51^V spectra are in good agreement with the recent literature [18,29]. The elemental ratio of P/Mo/V determined via ICP-OES was P/Mo/V = 1/6,70/4,76, with the Mo/V ratio being 7/5.

We chose the H_8_[PV_5_Mo_7_O_40_] (HPA-5) Keggin-type polyoxometalate as a catalyst for our heterogeneous system because this compound had already been successfully used for the homogeneous glucose to formic acid methyl ester oxidation, and it was reported to be more catalytically selective and more easily re-oxidized by molecular oxygen than less substituted phosphomolybdic acids [29,31]. In addition, the H_8_[PV_5_Mo_7_O_40_] (HPA-5) catalyst, like other polyoxometalates, is itself a strong inorganic acid with a catalytically active anion [32]. It fully dissociates in aqueous solution, which makes the immobilization of the catalyst onto the PS-*b*-PDMAEMA membrane possible via simple wet impregnation overnight (Figure 3). Based on this, the block copolymer membrane was modified with the catalyst by placing it into an aqueous HPA-5 solution overnight. 

The successful electrostatic attachment of the catalyst was observed as the membrane color turned from white/transparent to pale orange (Appendix A). The polyoxometalate uptake was measured via TGA and was in the range from 26 to 28 wt% (Figure 4), which corresponds to about 8.5 DMAEMA units per catalyst anion. Further increase of concentration during loading did not lead to an increase in the catalyst load. We tentatively assign the change of the slope of the TGA curve of a pristine membrane at around 420 °C to differences in thermal degradation of both blocks, while the change for a membrane after catalyst loading presumably occurs due to the presence of the HPA-5 catalyst electrostatically attached to the PDMAEMA block.

### 3.3. Catalytic Performance

Prior to heterogeneous catalytic experiments, three homogeneous glucose oxidations were carried out for future comparison at 11.5 bar air pressure in the OLT-HP-50 high-pressure reactor (Appendix A). Three reactions on the scale of 30.0 mg of glucose using 2.5 mol% of HPA-5 POM resulted in an average calculated yield of 55.7 ± 4.6%. Head-space GC–MS measurements for homogeneous reactions were performed using three technical replicates of each reaction mixture. The moderate yields obtained were expected, considering previous literature data [18]. In the work on the homogeneous glucose to formic acid methyl ester oxidation, the authors carried out the reactions by varying the initial oxygen partial pressure from 25 to 3 bar, obtaining the highest formic acid methyl ester yield of 95% at 25 bar and the lowest of 75% at 3 bar using 10 mol% of HPA-5 POM [18]. In our case, our lab setup allowed us to work at the maximum of 11.5 bar air pressure, which is equal to 2.4 bar oxygen partial pressure, at which moderate yields are expected. At the same time, the given oxygen partial pressure was sufficient for us to study the performance of our catalytic membranes over multiple consecutive cycles.

The catalytic activity of the PS-*b*-PDMAEMA block copolymer membrane modified with H_8_[PMo_7_V_5_O_40_] (HPA-5) toward glucose to formic acid methyl ester oxidation was tested at 11.5 bar air pressure in a high-pressure reactor at 90 °C for 24 h using 2.5 instead of 10 mol% of the HPA-5 catalyst. The general reaction scheme and the corresponding catalytic cycle for the HPA-5 catalyst are shown in Figure 5. The reactions were carried out in methanol. The choice of methanol is based on higher oxygen solubility compared to water, which allows to work at lower oxygen pressures and, at the same time, methanol enhances the HPA-5 catalyst selectivity [33]. The formic acid methyl ester yields were quantified via GC–MS using pentane as the internal standard; five technical replicates were collected after each catalytic cycle.

It was previously experimentally proved by the ^13^C isotopic labeling method that the formic acid methyl ester originates exclusively from glucose oxidation by HPA-5 POM and molecular oxygen, but not from the oxidation of methanol. The glucose oxidation to formic acid methyl ester is considered to begin with the oxidative cleavage of the C–C bond, leading to the formation of erythrose and glyoxal. Erythrose undergoes further C–C bond cleavage with the formation of glyoxal and glycolaldehyde. Two glyoxal and one glycolaldehyde molecule undergo C–C bond cleavage, forming formic acid molecules, which are subsequently converted into formic acid methyl ester via esterification with methanol (Figure 6). Overall, six formic acid methyl ester molecules are formed from one glucose molecule [18]. 

Next, three sets of heterogeneous catalytic oxidation cycles were carried out, and the yields were estimated via head-space GC–MS (Table 1; Figure 7). The reaction scales for sets 1, 2 and 3 were 36.5, 31.3 and 28.5 mg of glucose, respectively, where the actual mass of glucose used was adjusted to the weight of the respective membrane, so that the catalyst amount remained at the constant value of 2.5 mol%. The reactions were carried out under high oxygen pressure. An additional heterogeneous reaction with a pristine polymer membrane was carried out, resulting in zero formic acid methyl ester yield, eliminating the possibility of glucose oxidation for formic acid methyl ester in the absence of the HPA-5 catalyst. 

Overall, as can be seen in Table 1, the catalytically active membranes maintain catalytic activity for multiple cycles, resulting in high yields (at a given oxygen pressure) of formic acid methyl ester during the first two cycles and in moderate yields for cycles 3–5. For each set of cycles, a gradual decrease in catalytic activity was observed, which seemed more prominent in sets 1 and 3. Set 2 shows slightly elevated activity, while sets 1 and 3 are more comparable in terms of efficiency. It is worth mentioning that, for all three sets, the membrane color turns green after the first cycle, indicating incomplete re-oxidation of V(IV) to V(V) via molecular oxygen (Appendix A). In our opinion, this can be considered as one of the reasons for the decrease in catalytic efficiency, and, at the same time, the membranes retain their general mechanical stability over prolonged periods of reaction time (more than 120 h).

### 3.4. Catalytic Membrane Stability and Possibility of Reactivation

The morphology of the block copolymer membranes after catalysis was analyzed via scanning electron microscopy (SEM, Figure 8).

As can be seen in the micrographs, the membrane underwent no significant morphological changes over a prolonged reaction time under harsh conditions. For the catalytic cycles of sets 2 and 3, the quantification of P, Mo and V ions leaching after each consecutive cycle was performed by means of ICP (Appendix A, Table 2). The most important ion, however, is V(V), because it is responsible for the oxidation process. The leaching of ions in the [PMo_7_V_5_O_40_]^8−^ is not stoichiometric within each set, with vanadium loss being the highest. The latter might be explained by the reversible HPA-5 anion dissociation releasing a vanadyl VO_2_^+^ cation [33,34], suggesting that the obtained values are the combination of anion detachment from the pore surface and reversible anion dissociation. The values for ion leaching were different between the two sets for each consecutive cycle, while similar trends for vanadium and molybdenum ions were observed. The loss of vanadium was largest in the first two cycles of both sets, representing 44.5/17.7 and 39.6/17.2%, respectively; the overall vanadium loss was 87.2 and 79.4% for sets 2 and 3. In the case of molybdenum ions, the leaching was gradual for both cycles, representing 11% on average after each consecutive cycle in both sets. The loss of P was minor or negligible for both cases. These results emphasize that even after a significant leaching of around 80% of all vanadium ions, the catalyst still maintains nearly half of its initial catalytic activity.

In addition, the changes in the content of P, Mo and V in the pristine membrane and in the modified membrane before and after the catalytic process were analyzed via XPS (Figure 9). For comparison, the data from H_8_[PV_5_Mo_7_O_40_] XPS were also added. The signals in the XP spectrum of the H_8_[PV_5_Mo_7_O_40_] catalyst are a P 2p doublet (P-O) at 133.7/134.6 eV (energy separation 0.87 eV) assigned to P(V), a Mo 3d doublet (Mo-O) at 233.0/236.1eV (energy separation 3.15 eV) corresponding to Mo(VI) and a V 2p doublet (V-O) at 517.7/525.1 eV (energy separation 7.40 eV) corresponding to V(V). The successful electrostatic attachment of the heteropolyoxometalate anion was confirmed by the presence of its signal patterns in the spectrum of the modified membrane before catalysis. The content of P, Mo and V in the modified membrane was 0.6, 4.0 and 1.6 at%, respectively. It was also noticed that the V 2p doublet at 516.1/523.5 eV, belonging to the V(IV) ion, appeared after the catalyst immobilization on the membrane, with a quantitative ratio of V(V) to V(IV) of 4:1. In the XP spectrum of the membrane after five catalytic cycles, the P(V) 2p doublet completely disappeared; the Mo(VI) 3d doublet signal decreased, as the signal dropped from 4.0 to 0.5 at%; and for vanadium, the peak V(V) 2p was absent, and only a weak signal corresponding to V(IV) was recorded, with intensity drop from 1.6 to 0.1 at%. The presence of the V(IV) signal confirms the assumption of incomplete oxidation of the catalyst at 2.4 bar partial oxygen pressure. From our point of view, the analysis of the changes in the content of Mo, V and P in a catalytic membrane after catalytic reactions via XP spectroscopy is merely qualitative because of the small analysis depth of this technique (≈5–10 nm) in comparison to the thickness of the membrane (tens of µm). As a result of this, in the case of our material, we mostly rely on bulk analysis of the reaction solutions via the ICP-OES analysis technique for the quantification of catalyst leaching.

For set 1, an attempt to reactivate the membrane after the fifth catalytic cycle by immersing it again into an HPA-5 water solution was made, assuming that catalyst leaching might be the main source of catalytic efficiency loss. Thereafter, an additional sixth catalytic cycle was carried out, which resulted in further catalytic efficiency loss, with a formic acid methyl ester yield of only 16.0%. As this was somewhat unexpected, a part of a membrane from set 2 was taken for TGA analysis after the fifth cycle and after further immersion into the HPA-5 solution. The TGA before and after the immersion demonstrated no significant change in the inorganic material content in the membrane. This might have suggested that during the catalytic reaction, the anchoring groups (DMAEMA) on the pore surface underwent chemical degradation. For the moment, due to the poor solubility of the catalytic membranes in chloroform, tetrahydrofuran and dimethylformamide, we were not able to obtain meaningful results from ^1^H NMR spectroscopy. In our second approach, to determine whether DMAEMA units underwent chemical changes or not, the N 1s XP spectrum of a pristine polymer membrane and a modified membrane before and after catalysis was measured (Figure 10). The nitrogen in the pristine membrane is characterized by a peak at 399.6 eV corresponding to the non-protonated nitrogen in the DMAEMA repeating unit. Upon catalyst immobilization, around 85–90% of DMAEMA units become protonated, and an additional nitrogen signal shifted to higher binding energy at 402.6 eV appears. The exact quantification of the ratios is difficult due to Mo 3p_3/2_ signal overlapping (see Section 2.7 for details).

The last XP spectrum demonstrates that significant amounts of nitrogen were still present in the polymer membrane after multiple catalytic cycles and that the N 1s orbital did not experience any shifts to higher binding energies, indicating that the DMAEMA group was not oxidized. In contrast to the catalytic membrane before catalysis, only a negligible portion of DMAEMA groups were present in the protonated form, while most of them existed in a non-protonated form after the catalytic reaction. In conclusion, the results obtained indicate that DMAEMA anchoring groups remain chemically inert over long-term reaction in the harsh oxidative conditions.

In the end, having assumed that the main cause of unsuccessful membrane reactivation might be the hindering of active attachment sites by the products of chemical degradation of the HPA-5 catalyst, we decided to remove the remaining catalyst completely and then reactivate the membrane again via wet immersion, as described earlier. To achieve that, the membrane was placed into a brine solution under stirring. Successful removal could be observed by the slow change of the membrane color from green to white; complete catalyst detachment took around three weeks. After reactivation, the catalyst load estimated via TGA was 10 wt%.

## 4. Conclusions

Herein, we developed a block copolymer-supported heterogeneous catalyst for oxidative glucose to formic acid methyl ester conversion. The given reaction is of interest, as it opens the route for the conversion of biomass into a chemical widely used in industry. The catalyst design is based on the electrostatic anchoring of the inorganic HPA-5 POM H_8_[PV_5_Mo_7_O_40_] by positively charged DMAEMA units of the porous PS-*b*-PDMAEMA membrane fabricated via NIPS. It was demonstrated that the catalytic membrane retains its catalytic activity over multiple consecutive cycles of glucose oxidation, producing yields similar to homogeneous catalysis under identical conditions. The changes in the catalytic membrane properties—i.e., catalyst load, porosity, oxidation states of the elements in the inorganic catalyst—taking place during the prolonged reaction times were studied by applying different analytical techniques. According to the results obtained, the DMAEMA anchoring groups remain chemically inert during catalysis, while the gradual decrease in oxidative efficiency could at first be caused by catalyst leaching, and second, by insufficient partial oxygen pressure required for complete catalyst re-oxidation. The latter can be solved technically by increasing the oxygen partial pressure, probably leading to the increase in reaction yield in each consecutive cycle. Based on the results obtained, we can conclude that our system is well suitable for up to five consecutive glucose oxidation cycles without further reactivation.

## Figures and Tables

**Figure 1 nanomaterials-13-02498-f001:**
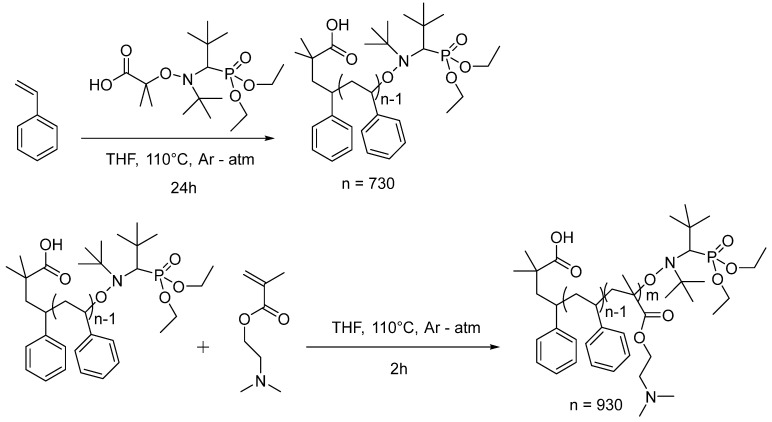
Two-step synthesis of the PS-*b*-PDMAEMA block copolymer via NMP.

**Figure 2 nanomaterials-13-02498-f002:**
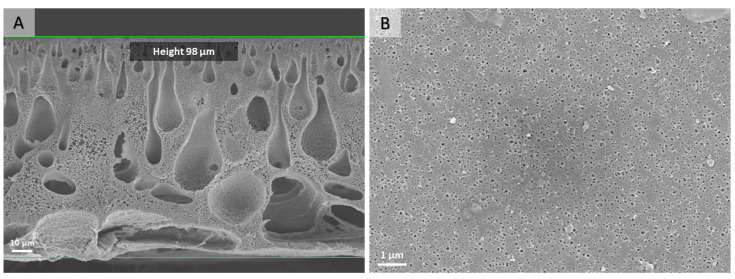
SEM micrographs of the cross-section (**A**) and top view (**B**) of a pristine PS-*b*-PDMAEMA membrane fabricated via the NIPS process.

**Figure 3 nanomaterials-13-02498-f003:**
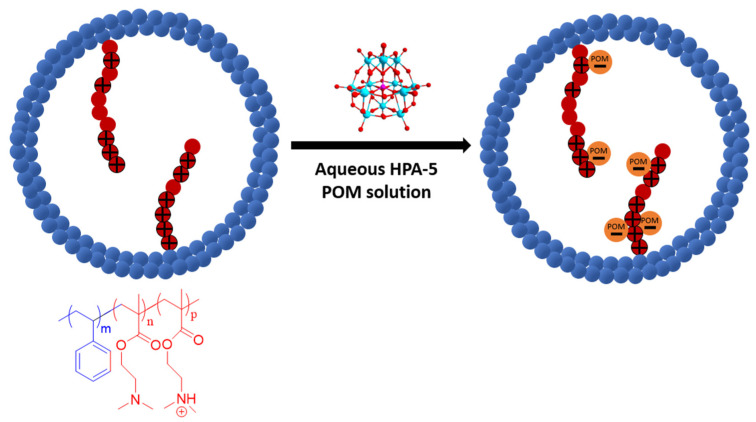
Schematic representation of the polyoxometalate immobilization onto the pore surface due to electrostatic interactions of positively charged DMAEMA groups and HPA-5 POM anions.

**Figure 4 nanomaterials-13-02498-f004:**
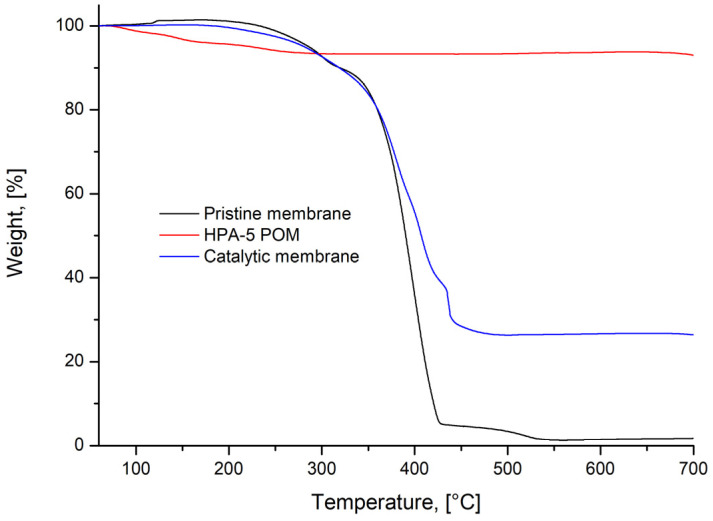
TGA profiles of the pristine PS-*b*-PDMAEMA membrane (black curve), HPA-5 POM catalyst (red curve) and PS-*b*-PDMAEMA membrane functionalized with HPA-5 POM catalyst (blue curve).

**Figure 5 nanomaterials-13-02498-f005:**
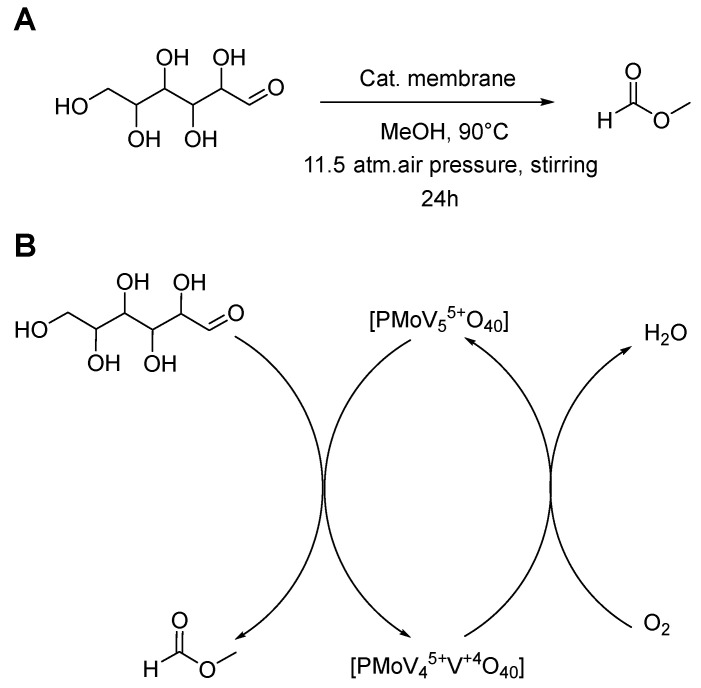
(**A**): Reaction conditions for the glucose to formic acid methyl ester oxidation in a high-pressure reactor; (**B**): Schematic representation of the proposed catalyst reduction-oxidation pathway involving molecular oxygen for the re-oxidation of the reduced form of the catalyst.

**Figure 6 nanomaterials-13-02498-f006:**
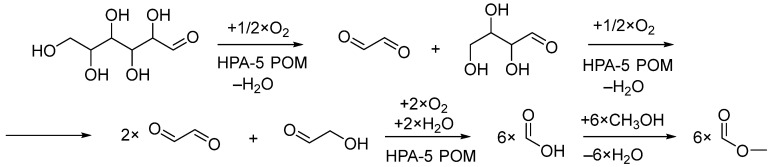
Catalytic pathway of the glucose to formic acid methyl ester oxidation [18].

**Figure 7 nanomaterials-13-02498-f007:**
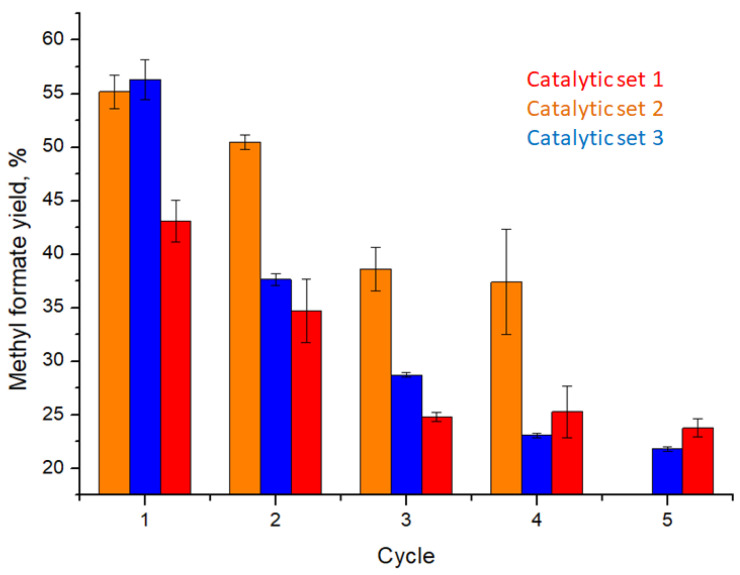
Histogram representation of the catalytic activity of polymer membranes modified with HPA-5 POM.

**Figure 8 nanomaterials-13-02498-f008:**
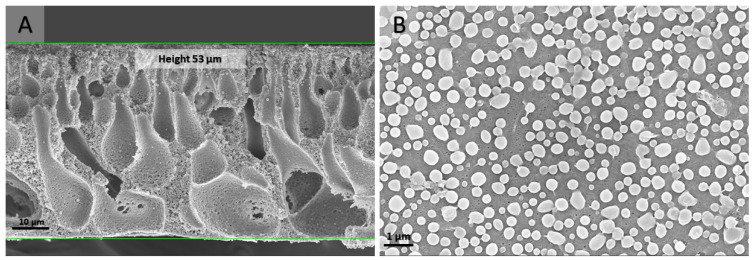
SEM micrographs of the cross-section (**A**) and top view (**B**) of the HPA-5 POM modified PS-*b*-PDMAEMA membrane after prolonged catalysis.

**Figure 9 nanomaterials-13-02498-f009:**
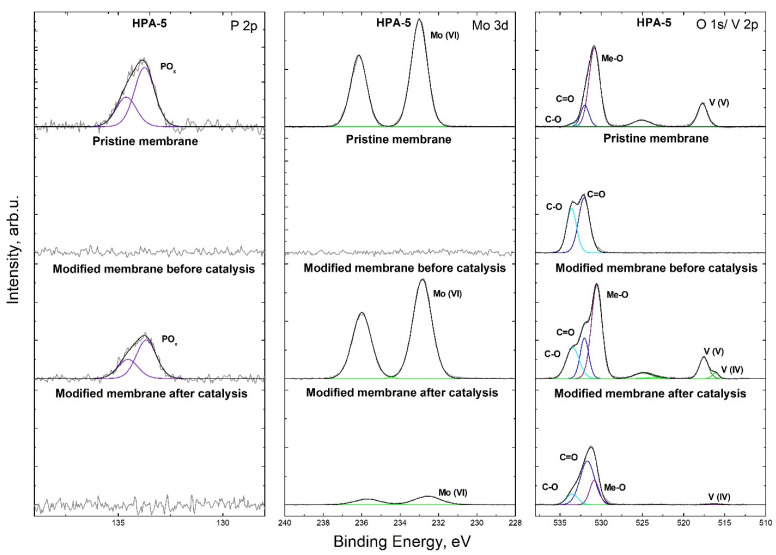
High-resolution P 2p (**left**), Mo 3d (**middle**) and O 1s/V 2p (**right**) XP spectra of the HPA-5 POM and the surface of pristine membrane/modified membrane before and after catalysis.

**Figure 10 nanomaterials-13-02498-f010:**
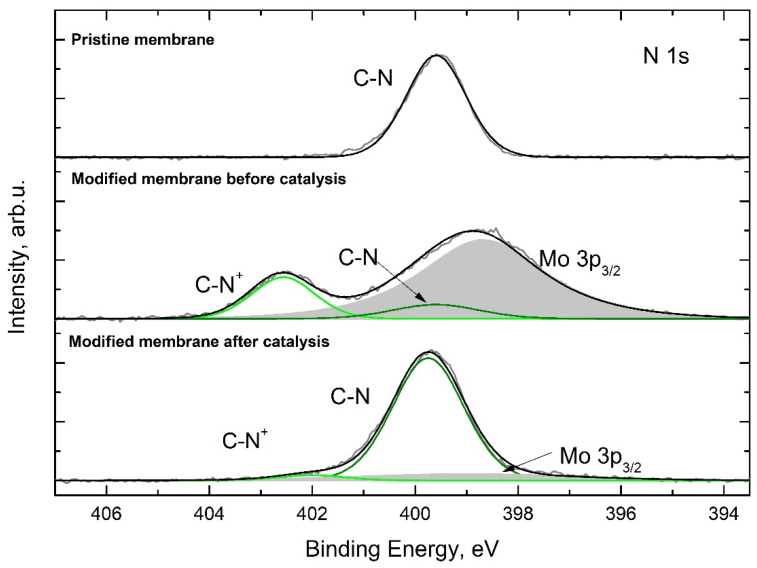
High-resolution N 1s XP spectra of the surface of pristine membrane/modified membrane before and after catalysis.

**Table 1 nanomaterials-13-02498-t001:** Catalytic activity of HPA-5 POM functionalized block copolymer membranes toward glucose to formic acid methyl ester oxidation in three catalytic sets calculated via head-space GC–MS. The confidence interval was calculated at 95% confidence level.

Catalytic set 1					
Cycle	1	2	3	4	5
Yield, %	43.1 ± 1.9	34.7 ± 2.9	24.8 ± 0.4	25.3 ± 2.4	23.8 ± 0.8
Catalytic set 2					
Cycle	1	2	3	4	5
Yield, %	55.2 ± 1.6	50.5 ± 0.7	38.6 ± 2.0	37.4 ± 4.9	-
Catalytic set 3					
Cycle	1	2	3	4	5
Yield, %	56.3 ± 1.9	37.7 ± 0.6	28.7 ± 0.2	23.1 ± 0.2	21.8 ± 0.2

**Table 2 nanomaterials-13-02498-t002:** Catalyst leaching after each cycle from catalytic sets 2 and 3 calculated via ICP-OES.

	Catalytic Set 2			Catalytic Set 3		
Element	Mo Loss, %	V Loss, %	P Loss, %	Mo Loss, %	V Loss, %	P Loss, %
Cycle 1	13.0	44.5	13.7	11.2	39.6	0
Cycle 2	12.3	17.7	10.5	9.9	17.2	0
Cycle 3	11.7	10.6	0	9.9	9.7	0
Cycle 4	11.9	7.6	0	10.4	7.1	0
Cycle 5	12.4	6.9	0	11.6	5.9	0
Overall Loss	61.3	87.2	24.2	53.1	79.4	0

## Data Availability

Data are available on reasonable request from the corresponding author.

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
