# Peer review of "Polyoxometalate-Modified Amphiphilic Polystyrene-block-poly(2-(dimethylamino)ethyl methacrylate) Membranes for Heterogeneous Glucose to Formic Acid Methyl Ester Oxidation"

_nanomaterials, 2023, doi:10.3390/nano13182498_

Round 1
Reviewer 1 Report
The authors have presented an approach towards electrostatic immobilization of HPA-5 POM within the pores of amphiphilic PS-b-PDMAEMA block copolymer membranes, which was characterized and tested for multiple consecutive cycles to catalyze glucose to formic acid methyl ester oxidation reactions. However, some of the following points, in my opinion, could be further improved or clarified. It may be reconsidered for publication after major revision.
1. They reported that the membrane casting under chosen conditions resulted in a microporous volume structure and a more fine-porous surface layer, however, the porous volume was not measured.
2. The authors compared their study with reference [18], however, the oxygen partial pressure and concentration of HPA-5 were both different, which makes the reader confused.
3. In homogeneous glucose oxidation process, the average calculated yield reached 55.7
4.6%, which is higher than that in heterogeneous oxidation process. Also, the reaction temperature was set at 90 ℃ in heterogeneous oxidation process. Obviously, the heterogeneous catalytic process is lack of advantage for glucose oxidation.
4. They pointed out that even after a significant leaching of around 80% of all vanadium ions the catalyst still maintains nearly half of its initial catalytic activity. However, the leached vanadium ions may be presented in the liquid phase, which also obtained catalytic activity. Furthermore, the quantification of Mo and V ions leaching after each consecutive cycle is huge, which indicates that the loading of HPA-5 on membranes is unsteady.
5. The methyl formate yield in three Catalytic set is significantly different. Why?
Moderate editing of English language required
Author Response
We thank all reviewers for their constructive criticism. Please find point-to-point answers below. All changes in the revised manuscript have been highlighted in yellow.
Reviewer #1:
The authors have presented an approach towards electrostatic immobilization of HPA-5 POM within the pores of amphiphilic PS-b-PDMAEMA block copolymer membranes, which was characterized and tested for multiple consecutive cycles to catalyze glucose to formic acid methyl ester oxidation reactions. However, some of the following points, in my opinion, could be further improved or clarified. It may be reconsidered for publication after major revision.
- They reported that the membrane casting under chosen conditions resulted in a microporous volume structure and a more fine-porous surface layer, however, the porous volume was not measured.
Thanks for the hint. The total pore volume as well as surface specific area were determined after your remark and can be found in the paragraph «Preparation of catalytically active membranes» in the «Results and Discussion section». The plots can be found in the ESI as Figures S3 and S4.
- The authors compared their study with reference [18], however, the oxygen partial pressure and concentration of HPA-5 were both different, which makes the reader confused.
Thanks for your question. Indeed, in our work we applied different oxygen partial and used different mol% of the catalyst compared to the reference [18]. It is also worth mentioning that both oxygen partial pressure and mol% of the HPA-5 catalyst in our work were lower than in [18] where the homogeneous glucose oxidation was established. Answering your question generally, in our work we did not intend to work in the same range of oxygen partial pressure (3 – 20 bar pure) and HPA-5 mol% (10 % mol) as it was in article [18]. In the presented work we applied 2.4 bar of oxygen partial pressure (11.5 bar of air) because from our considerations it would be totally enough (and safe) to test our heterogeneous system for catalytic activity and show the properties of the polymeric membranes. The difference in catalyst load is due to the amount of membrane we could produce per one preparation procedure. We wanted one catalytic set to be performed with membranes from one batch, without mixing different batches. From one preparation procedure (batch) in our conditions we could get around 20 mg of material. With this amount and the desired amount of glucose we wanted to use for oxidation we could not go beyond 2.5% mol HPA-5. So, overall, reference [18] serves as a general comparison and, in general, not so many examples for this type of heterogeneous catalysis have been reported.
- In homogeneous glucose oxidation process, the average calculated yield reached 55.74.6%, which is higher than that in heterogeneous oxidation process. Also, the reaction temperature was set at 90 ℃ in heterogeneous oxidation process. Obviously, the heterogeneous catalytic process lack of advantage for glucose oxidation.
For both homogeneous oxidation (single oxidation, without consecutive cycles) and consecutive heterogeneous oxidations the reaction temperature as well as air pressure were identical, 90°C and 11.5 bar respectively. From Table 1 we can see that for Catalytic sets 1, 2 and 3 the yield of formic acid methyl ester in the first cycle were 43.1, 55.2 and 56.3 % respectively, compared to average 55.7 % yield for homogeneous oxidation. This result proved that the HPA-5 loaded onto the membrane has on average same/nearly the same catalytic activity as free HPA-5 in the MeOH solution, which at the beginning was not guaranteed.
In the present work only comparison of the homogeneous oxidation yield with the first cycle yield of heterogeneous oxidations would be correct. At the same time, it is guaranteed that free HPA-5 would also lose its activity over time because at our working oxygen pressure V(+4) cannot be fully reoxidized to V(+5).
To make a general comparison, we would have to carry out consecutive homogeneous oxidations, which are technically not feasible for us as they would result in difficulties for product calculation via Head-space GC. The reason for this is given in the next paragraph.
From experimental point of view, the use of catalytic membranes as a heterogeneous catalyst allows us to completely remove old reaction solution after each cycle without removing the catalyst, and then add a fresh methanol/glucose/pentane solution for the next cycle. This procedure would be experimentally impossible in case when the HPA-5 is simply dissolved in methanol. (When the catalyst is heterogeneous, we remove the methanol solution but the membrane remains inside the reaction vessel. Contrary, when the HPA-5 is simply dissolved in MeOH we can’t remove MeOH without removing the catalyst). The complete exchange of the reaction solution for a newly prepared one is important for precise yield quantification via Headspace-GC, which requires exactly the same and known amount of internal standard (pentane) for each cycle and, besides that, is a general advantage for heterogeneous catalysis.
Another advantage heterogeneous setup in the end after the 5-th cycle we managed to keep around 40% of Molybdenum and around 20% of vanadium on our membrane, preventing it from leaching to a certain extent – which might be beneficial considering the environmental aspects of such a process.
- They pointed out that even after a significant leaching of around 80% of all vanadium ions the catalyst still maintains nearly half of its initial catalytic activity. However, the leached vanadium ions may be presented in the liquid phase, which also obtained catalytic activity. Furthermore, the quantification of Mo and V ions leaching after each consecutive cycle is huge, which indicates that the loading of HPA-5 on membranes is unsteady.
After each consecutive cycle we completely removed the reaction solution and washed the membrane twice with fresh methanol. After that a new portion of methanol, glucose and pentane was added for the next cycle (the procedure can be found in the Materials and Methods section, «Catalytic heterogeneous glucose to formic acid oxidation» paragraph). Due to this there is no possibility for accumulation of leaching catalyst after each consecutive catalytic cycle.
In our work, the HPA-5 catalyst is being held on the surface of the polymeric membrane due to its electrostatic interactions with the tertiary amino group of the PDMAEMA block (which is exposed to the membrane surface). From the results of the ICP-OES measurement, we could observe a noticeable vanadium leaching only after the first catalytic cycle, while during the next cycles leaching decreased. In this work we could observe that the factor of elevated temperature might constitute a certain complications for electrostatic catalyst attachment, in comparison to reactions that can be carried at room temperature. Nevertheless the catalyst maintained nearly half of its initial catalytic activity after in total 120 hours of reaction time. In the future, strategies might be of advantage where a partially covalent anchorage of the catalyst can be targeted.
- The methyl formate yield in three Catalytic set is significantly different. Why?
We assume that the differences in yield represent the sum of all experimental errors, i.e. heating (to the desired temperature) and cooling time of the reactor, sampling, GC-measurements, possible slight variations in membrane porosity as well as slight differences in the reaction scale as we had to adjust the overall amount of glucose in each case to the actual amount of membrane used (and with that and similar loadings the overall amount of catalyst).
Reviewer 2 Report
This article is well organized and provides clear results.
Although it is a little thing, if you may have some comments on the bump around 420C shown in Figure 4 TGA data, it would be better.
Author Response
Reviewer #2:
This article is well organized and provides clear results.
Although it is a little thing, if you may have some comments on the bump around 420C shown in Figure 4 TGA data, it would be better.
Thank you for your feedback. There is indeed a systematic bump at 420 degrees for catalytic membranes. We have added the explanation of this affect in the paragraph above the TGA plots.
Reviewer 3 Report
The manuscript describes the fabrication of Polyoxometalate-modified amphiphilic polystyrene-block-poly(2-(dimethylamino)ethyl methacrylate) membranes for heterogeneous glucose to formic acid methyl ester oxidation. The obtained membrane was characterized and applied for catalytic reaction adequately. Thus, it can be accepted after addressing the following minor issues.
The abstract can be improved by stating the significant findings & results.
The introduction is too long. rewrite it in three concise paragraphs.
The active surface area (i.e. exposed area) of the membrane needs to be added in the catalytic part.
Include the ICP-OES plots in ESI.
Table 2 also can be compared with the At.% of elements calculated from XPS before and after the catalytic reaction.
Author Response
Reviewer #3:
The manuscript describes the fabrication of Polyoxometalate-modified amphiphilic polystyrene-block-poly(2-(dimethylamino)ethyl methacrylate) membranes for heterogeneous glucose to formic acid methyl ester oxidation. The obtained membrane was characterized and applied for catalytic reaction adequately. Thus, it can be accepted after addressing the following minor issues.
The abstract can be improved by stating the significant findings & results.
We added some improvement to the abstract
The introduction is too long. rewrite it in three concise paragraphs.
We shortened some parts of the introduction.
The active surface area (i.e. exposed area) of the membrane needs to be added in the catalytic part.
The total pore volume as well as surface specific area were determined after your remark and can be found in the paragraph «Preparation of catalytically active membranes» in the «Results and Discussion section». The plots can be found in the ESI as figures S3 and S4.
Include the ICP-OES plots in ESI.
The ICP-OES plots of Mo, V and P were added upon your request to the ESI, Figures S10-15.
Table 2 also can be compared with the At.% of elements calculated from XPS before and after the catalytic reaction.
We added the corresponding explanation about the difference in reliability of these two techniques for leaching quantification in our work.
Round 2
Reviewer 1 Report
Accept in present form.